# HLA-G and Other Immune Checkpoint Molecules as Targets for Novel Combined Immunotherapies

**DOI:** 10.3390/ijms23062925

**Published:** 2022-03-08

**Authors:** Fabio Morandi, Irma Airoldi

**Affiliations:** Stem Cell Laboratory and Cell Therapy Center, IRCCS Istituto Giannina Gaslini, 16148 Genova, Italy; irmaairoldi@gaslini.org

**Keywords:** HLA-G, immune checkpoints, tumors

## Abstract

HLA-G is an HLA-class Ib molecule that is involved in the establishment of tolerance at the maternal/fetal interface during pregnancy. The expression of HLA-G is highly restricted in adults, but the de novo expression of this molecule may be observed in different hematological and solid tumors and is related to cancer progression. Indeed, tumor cells expressing high levels of HLA-G are able to suppress anti-tumor responses, thus escaping from the control of the immune system. HLA-G has been proposed as an immune checkpoint (IC) molecule due to its crucial role in tumor progression, immune escape, and metastatic spread. We here review data available in the literature in which the interaction between HLA-G and other IC molecules is reported, in particular PD-1, CTLA-4, and TIM-3, but also IDO and TIGIT. Clinical trials using monoclonal antibodies against HLA-G and other IC are currently ongoing with cancer patients where antibodies and inhibitors of PD-1 and CTLA-4 showed encouraging results. With this background, we may envisage that combined therapies using antibodies targeting HLA-G and another IC may be successful for clinical purposes. Indeed, such immunotherapeutic protocols may achieve a better rescue of effective anti-tumor immune response, thus improving the clinical outcome of patients.

## 1. Introduction

HLA-G is a non-classical MHC class I molecule with immune-suppressive properties. It belongs to the group of HLA-class Ib molecules that also includes HLA-E, -F, and -H. Among these, HLA-G is the most characterized, and studies addressed its role in physiological and pathological conditions. In contrast with HLA-class Ia molecules (HLA-A, -B, and –C), that are involved in antigen recognition through the interaction with T-cell receptor/peptide complexes, the main role of HLA-class Ib molecules is the modulation of immune responses. 

HLA-G gene maps in the short arm of chromosome 6 at region 6p21.3, within the class I gene cluster of major histocompatibility complex (MHC). The primary transcript may generate at least seven different isoforms through alternative splicing, four of which retain the transmembrane domain and, for this reason, are expressed on the cell surface (i.e., HLA-G1, -G2, -G3, and –G4), whereas the other three isoforms display intronic sequences and lack transmembrane and cytoplasmic domains. Thus, these latter forms that include HLA-G5, -G6, and –G7 are present as soluble (s) moieties. These different isoforms are also characterized by a different number of globular domains [1]. HLA-G1 and -G5 are the most studied and display an identical structure, meaning that they are composed of a heavy chain (HC) with three α-globulins. By contrast, HLA-G2 (which share an identical structure with soluble HLA-G6) and HLA-G4 are composed by only two globular domains, and HLA-G3 and -G7 display only one globular domain. All HLA-G molecules are described as free-HC [2] or as heterotrimers composed by HC non-covalently bound to β2-microglobulin and to peptides (nonamers) generated from the proteasomal degradation of different intracellular proteins [3]. 

HLA-G has been also described in the last years on the membrane of extracellular vesicles (EV) that play an important role in the modulation of immune response [4,5] either in physiological [6,7] or pathological [8,9] conditions.

HLA-G molecules interact with different receptors, the most important of which are the immunoglobulin-like transcript (ILT)2 and ILT4 [10,11] found on a wide variety of immune cells of both myeloid and lymphoid origin. In particular, ILT2 mainly recognizes HLA-G1 and -G5 associated with β2M and is expressed on monocytes, B and T lymphocytes, NK and dendritic cells (DC), as well as on myeloid-derived suppressive cells (MDSCs) [12]. On the other hand, ILT4 recognizes the HLA-G2 and HLA-G6 isoforms but also β2M-free HLA-G1 and HLA-G5, and it is mainly present on monocytes, neutrophils, DC, and MDSCs [13,14,15]. Both ILT2 and ILT4 display four extracellular immunoglobulin domains and four and three immunoreceptor tyrosine-based inhibitory receptor motifs (ITIM), respectively, in their cytoplasmic tails. In turn, ITIM recruit the protein tyrosine phosphatase SHP-1 (Src homology 2 domain containing phosphatase 1), which de-phosphorylate signaling proteins involved in early events triggered by stimulatory receptors [16]. Another HLA-G receptor is KIR2DL4, which is expressed on NK cells and a subset of T lymphocytes; it recognizes all HLA-G isoforms through the α1 domain [17,18,19]. KIR2DL4 contains only one intracellular ITIM motif, and it may interact with ITAM motifs containing adaptor DAP12 through the interaction with a charged residue in the transmembrane domain. Thus, upon the interaction with HLA-G, KIR2DL4 may deliver both activating and inhibitory signals [20].

As mentioned, the immune suppressive role of HLA-G has been clearly established in some physiological conditions, the most intriguing of which is the maternal/fetal interface on extravillous cytotrophoblast. During pregnancy, such semi-allogenic fetal tissue invades the maternal uterus without being rejected by the maternal immune system, which is a feature that is an apparent violation of the laws of transplantation. So, the master molecule underlying this feature is HLA-G, which plays a crucial role in the establishment of immune tolerance by suppressing maternal immune response against fetal tissue [21]. In particular, placental cells synthetize both membrane and sHLA-G isoforms, which interact with ILT2 and ILT4 on different target cells, operating by multiple pathways including: (i) induction of death of cytotoxic T lymphocytes and down-regulation of their major co-receptor/activator cell surface molecule (i.e., CD8), (ii) immobilization of NK cells, and (iii) programming of phagocytes into suppressive modes by the production of anti-inflammatory cytokines. In adults, HLA-G expression is confined to a few tissues such as (i) the cornea [22], where it contributes to the maintenance of a privileged immune status, (ii) erythroblasts [23], where it is involved in differentiation and angiogenesis, and (iii) pancreatic islets [24], where it exerts regulatory functions with important implications for the progression of autoimmunity as well as for the establishment of transplant tolerance. In addition, HLA-G is expressed by mesenchymal stem cells [25] and regulatory T cells [26] and plays an important role for the immune-regulatory properties of these two cell populations. Finally, HLA-G expression is down-regulated in different autoimmune and inflammatory diseases and induced or up-regulated in viral and microbial infections [27,28] and in cancer, where it may be considered a tumor-associated antigen [29,30,31].

In the context of pathological conditions, cancer represents one of the major focuses of interest, and the role of HLA-G has been widely analyzed and studied. In hematological malignancies, especially in B cell tumors such as multiple myeloma [32], non-Hodgkin B-lymphoma, and B-CLL [12,33], HLA-G may exert anti-tumor activities through the direct inhibition of tumor cell proliferation. However, although an enhanced sHLA-G (i.e., HLA-G5 and shed HLA-G1) plasma levels has been observed, no clear correlation was established between HLA-G and unfavorable clinical outcome in these tumors [12,32,33]. Naji et al. [34] demonstrated that the proliferation of hematological tumors expressing ILT2 may be inhibited by HLA-G and that B cell proliferation can be restored by blocking HLA-G or its receptor ILT2 by specific antibody or siRNA. In addition, the reduction in B cell proliferation by HLA-G was dependent on G0/G1 cell cycle arrest and mTOR signaling blockade. The evidence that HLA-G may inhibit tumor cell proliferation was consolidated by studies using bone marrow (BM) samples from multiple myeloma patients where HLA-G limits CD138^−^ stem cell differentiation into CD138^+^ multiple myeloma cells [34].

Furthermore, we provided insights in the mechanisms of tumor suppression in vivo in this compartment, which was essentially mediated by the production of soluble HLA-G proteins by BM-derived mesenchymal stem cells [35] and by osteoblasts [36]. Nonetheless, it is important to mention that an opposite role of HLA-G has been also reported in hematological tumors. For example, HLA-G expression is altered in non-Hodgkin B-lymphoma and correlates to tumor relapse and transformation [31,37], but this feature has been postulated to be associated to the deep genetic disorder and rearrangement that, in turn, induces HLA-G neo-expression.

In human solid tumors, HLA-G was found to be expressed in a wide variety of tumor cells [27] such as colorectal [38], renal [39], lung [40], melanoma [41], pancreatic [42], hepatocellular [43], breast cancer [44], gastric carcinomas [45], and neuroblastoma [46,47]. In addition, HLA-G is expressed on tumor-infiltrating cells, especially on lymphocytes, monocytes, macrophages, and DC [41,48,49,50], and its clinical relevance is undisputed due to the following considerations. First, HLA-G expression is restricted to malignant transformation, but it is not present in the surrounding normal tissues [12]; second, the expression of HLA-G in biopsies and high levels of soluble HLA-G in plasma from neoplastic patients significantly correlates with poor prognosis [38,46,51,52,53]; third, HLA-G is found in solid tumors of high histological grades and advanced clinical stages [45,47,54].

Furthermore, HLA-G is involved in the relationship between immune system, tumor cells, and their microenvironment and in all three stages of tumor progression [30,55] (i.e., elimination, equilibrium, and escape), as described in the Schreiber’s theory [56]. In this scenario, HLA-G is usually up-regulated and produced by tumor cells and tumor-infiltrating leucocytes. HLA-G functions with paralleled mechanisms, through the ILT2/ILT4 signaling, that involve the inhibition of (i) proliferation of T and B lymphocytes, (ii) cytotoxicity of NK and T cells, (iii) phagocytic activity of neutrophils, and (iv) function of DC [57,58,59,60]. Furthermore, HLA-G dampens the secretion of pro-inflammatory cytokines and co-operates with other immunosuppressive and angiogenic molecules (e.g., IL-10, TGF-β1, and VEGF) in the generation of a tolerogenic microenvironment by promoting the expansion regulatory T cells and the secretion anti-inflammatory cytokines [30]. Thus, in the end, HLA-G contributes to the metastatic spread and to the evolution of aggressive tumor cells in different solid and hematological tumors [12,30,61].

Taken together, all these features highlight the role of HLA-G as an immune checkpoint (IC) molecule and as a novel therapeutic target for cancer immunotherapy, as recently discussed [62]. Notably, the substantial lack of expression of HLA-G in normal tissues, associated with its presence on tumor cells, makes HLA-G an ideal tumor-specific target to be used also for chimeric antigen receptor (CAR) development [63].

We here review data highlighting the correlation and/or interactions between HLA-G and other IC molecules, namely PD-1/PD-L1, CTLA-4, TIM-3, TIGIT, and IDO, in human tumors. On the basis of these observations, we discuss future possible therapeutic strategies based on the combination of (i) blocking antibodies against HLA-G or its receptors and (ii) blocking antibodies specific for other IC molecules, with the aim of fully restoring anti-tumor immune response.

## 2. Immune Checkpoint Molecules

IC molecules are generally expressed on the surface of T lymphocytes, where they regulate the physiological immune response following the recognition of specific ligands, thus maintaining homeostasis and self-tolerance. However, the expression of IC molecules is greatly increased on T lymphocytes infiltrating the tumor microenvironment, where IC-specific ligands are expressed by regulatory cell subsets, such as regulatory T cells, MDSC, tumor-associated macrophages, as well as by tumor cells themselves. Thus, tumor-infiltrating lymphocytes become exhausted or tolerized, thus becoming unable to eliminate tumor cells [64]. The most studied IC molecules are the co-inhibitory molecules programmed cell death receptor (PD)-1 (and its ligands PD-L1/2) and cytotoxic T lymphocytes-associated protein (CTLA)-4, T-cell immunoglobulin and mucin-domain containing (TIM)-3, T cell immune-receptor with Ig and ITIM domains (TIGIT), and indolamine dioxygenase (IDO) [64].

### 2.1. HLA-G and PD-1/PD-L1 Axis

PD-1 is a membrane protein belonging to the CD28 family and represents a key regulator of normal host physiology and of programmed cell death in lymphocytes [65]. This is expressed upon the activation on different T and B cell subsets, NK and some myeloid cells, and cancer cells [64,66], and its critical role in the maintenance of peripheral tolerance has been unambiguously demonstrated [67]. PD-1 has two different ligands that are PD-L1, which is expressed on a wide variety of cells homing in primary and secondary lymphoid organs, and PD-L2 with a more restricted expression on antigen-presenting cells [66]. PD-1/PDL-1 interaction limits T cell activation signals by inhibiting proliferation, survival, and cytokine release, thus representing one of the most characterized IC also investigated as an immunotherapeutic target.

Schwich et al. demonstrated that HLA-G may up-regulate the expression of PD-1 on T lymphocytes [68]. In detail, the interaction of sHLA-G with ILT2 on T cells leads to the up-regulation of the IC molecules PD-1, CTLA-4, and TIM-3 on ILT2^+^ CD8^+^ T lymphocytes but not on ILT2^−^CD8^+^ or CD4^+^ T cells. In contrast, the priming of T cells with HLA-G bearing EV resulted in an increased expression of other IC molecules only on ILT2^−^ CD8^+^ T cells. These results indicate that HLA-G may act on different T cell populations depending on whether it is delivered by EV or is present in free soluble form, thus functioning with complementary immunosuppressive pathways. Furthermore, the HLA-G-driven up-regulation of PD-1 on T lymphocytes may render them anergic in the presence of tumor cells expressing PD-L1/L2 [68]. In this line, Ullah et al. have demonstrated that differentiation stimuli may increase the expression of HLA-G and PD-L1 on cancer cells [69]. In particular, ovarian and gastric cancer cell lines, after differentiation to adipocyte or neurocyte, showed a reduced expression of the CD90 stem cell marker, thus decreasing the percentage of cancer stem cells (CSC), and an increased expression of both HLA-G and PD-L1. These data suggested that differentiating agents, although reducing CSC, which are resistant to chemotherapy, may worsen the clinical outcome of treated patients by creating an immune-suppressive circuit in the tumor microenvironment. This is supported by the finding that the high levels of HLA-G observed in the tumor microenvironment, induced by the differentiation of cancer cell lines, may increase the expression of PD-1 on T lymphocytes, which may be, in turn, suppressed by cancer cells expressing high levels of PD-L1 [69]. Chen et al. evaluated the expression of PD-1/PD-L1 and HLA-G/ILT-2/ILT-4 in patients with colorectal cancer (CRC) [70]. Such analysis was performed by flow cytometry on single-cell suspension from frozen CRC lesions, gating on EpCAM^+^ tumor cells. First, the authors demonstrated that CRC patients with distant lymph node metastasis were significantly associated with high levels of HLA-G, ILT-2, ILT-4, and PD-L1, thus suggesting that all these molecules are related to a worse prognosis. Indeed, patients with high expression, compared to those with low expression, of all these molecules display a lower overall survival. This study confirmed that PD-1 and HLA-G and their receptors are closely related in cancer patients and that the high expression of these molecules may be predictive of a worse prognosis [70]. A paired up-regulation of HLA-G and PD-L1 and/or PD-L2 was found also in pancreatic cancer [71], papillary thyroid cancer [72], oral osteosarcomas [73], oral precancerous lesions [74], intraoral mucoepidermoid carcinomas [75], and adenoid cystic carcinomas of salivary glands [76], thus supporting the concept that the close interaction of HLA-G with PD-1 takes place in different human tumors to drive a successful evasion from the control of the immune system.

In a recent study, Dumont et al. analyzed TIL in renal cell carcinoma patients [77]. They found that CD8^+^ILT2^+^ peripheral blood (PB) cells displayed an up-regulation of effector-related genes and effector-regulated genes, such as granzyme B and IL-7. Surface molecules involved in cytotoxic functions, such as NK cell receptors KIR2DL3, KIR2DL1, KIR3DL2, and NKp46, were also up-regulated in this subset, whereas co-stimulatory molecules such as CD28 and CD40 ligand were down-regulated. Accordingly, they confirmed, by flow cytometric analysis, that CD8^+^ILT2^+^ PB cells expressed low levels of CD28, CD27, and CD127, and highly expressed KLRG1, perforin, and granzyme B. CD8^+^ILT2^+^ cells were able to release cytotoxic granules and IFN-γ in response to anti-CD3 coated target cells, and such functions were blocked when target cells expressed HLA-G1, but they were restored in the presence of an anti-ILT2 blocking antibody. The authors demonstrated that among TILs, ILT2 and PD-1 expression was mutually exclusive, thus identifying two distinct subsets (i.e., CD8^+^ILT2^+^PD-1^−^ cells and CD8+ILT2-PD-1+ cells). CD8^+^ILT2^+^PD-1^−^ cells exhibited an effector phenotype, whereas the CD8^+^ILT2^−^PD-1^+^ counterparts display an effector memory phenotype, low expression of co-stimulatory molecules and cytotoxic granules, and high expression of exhaustion markers such as Tim-3, CD38, and CD69. Furthermore, CD8^+^ILT2^+^PD-1^−^ TILs show a higher cytotoxicity and IFN-γ release, in response to anti-CD3 coated target cells, compared to CD8^+^ILT2^−^PD-1^+^ cells. Again, such functions were blocked by HLA-G1 expressing target cells only in CD8^+^ILT2^+^PD-1^−^ TILs, and they were restored in the presence of anti-ILT2 blocking antibody. Collectively, these data demonstrated that the function of HLA-G/ILT2 and PD-1/PD-L1 checkpoint in renal cell carcinoma is dicotomic, and HLA-G^+^/PD-L1^+^ tumor cells may abrogate the function of two distinct cytotoxic effector cell subsets in the tumor microenvironment. Thus, a therapeutic strategy aimed at blocking both immune checkpoints may be more effective to restore anti-tumor immune responses.

All these data regarding the interactions between HLA-G and PD-1/PD-L1 are summarized in Figure 1. Furthermore, in the same figure, we describe the potential effects that may be obtained using anti-PD-1/PD-L1 plus anti-HLA-G antibodies. In detail, the blocking of PD-1/PD-L1 interaction, in association with that of HLA-G/ILT2, is able to rescue anti-tumor CTL functions. In addition, since HLA-G, either soluble or present on the surface of tumor-derived EV, is able to up-regulate PD-1 expression on T lymphocytes upon interaction with ILT2, it is tempting to speculate that the inhibition of this pathway may restore T cell cytotoxic activities against tumor cells.

To date, two phase I clinical studies are in progress (www.clinicaltrials.gov accessed on 30 January 2022) in cancer patients (Table 1). In the clinical trial #NCT04485013, 200 patients with different types of cancer including head and neck squamous carcinomas, non-small cell lung cancer, colorectal cancer, and triple negative breast cancer are currently recruited for the treatment with the full human monoclonal antibody against HLA-G, named TTX-080. Interestingly, the antibody is tested alone or in combination with the PD-1 inhibitor pembrolizumab or the EGF-R inhibitor cetuximab. The results will clarify the pharmacokinetics and the immunogenicity of TTX-080 and the preliminary efficacy of TTX-080 as monotherapy or in combination with pembrolizumab or cetuximab. In the other clinical trial (#NCT04991740), 140 patients with advanced solid tumors are recruited for the treatment with a bi-specific antibody against HLA-G and CD3 (JNJ-78306358) and will be evaluated for pharmacokinetics, immunogenicity, and efficacy in terms of disease evaluation. Thus, we are confident that in the future, novel therapeutic protocols will be designed based on the combination of anti-HLA-G antibodies with anti-PD-1 (pembrolizumab, nivolumab) or anti-PD-L1 (avelumab, atezolizumab, durvalumab) antibodies, or inhibitors, in order to improve the clinical outcome of cancer patients.

### 2.2. CTLA-4 and TIM-3

Cytotoxic T lymphocytes-associated protein (CTLA)-4 is an inhibitory receptor that negatively regulates T cell responses by reducing T cell proliferation and cytokine production, thus avoiding physiologically the generation of auto-reactive T lymphocytes and aberrant T cell responses [78,79]. CTLA-4, belonging to the immunoglobulin super-family, is structurally homologue to CD28 and binds to the same ligands that are CD80 and CD86, although with higher affinity and avidity [79].

CTLA-4 represents one of the first IC studied [64,80] that is neo-expressed in many adult and pediatric cancers or with increased expression in different tumors, mostly correlated with HLA-G and PD-L1/L2 [72,76]. The expression of CTLA-4, together with the other IC T-cell immunoglobulin and mucin-domain containing (TIM)-3, may be also increased on T lymphocytes after treatment with sHLA-G or EV bearing HLA-G, as observed for PD-1 [68].

TIM-3, similarly to CTLA-4, belongs to the immunoglobulin superfamily and may exert both inhibitory and co-stimulatory functions in physiological situations, but in the context of cancer, its role in the suppression of anti-tumor responses has been clearly defined [64,81].

Gupta et al. have demonstrated that on glioma cells [82], IL-1β increased the expression of HLA-G and HMGB1, a ligand of TIM-3, that in turn increases HLA-G on tumor cells upon interaction with another specific receptor, TLR-4 [82]. Thus, it is tempting to speculate that an immune-suppressive loop may be triggered by HMGB1, which is over-expressed in different human tumors and correlates with tumor progression [83,84] through the induction of HLA-G expression upon the interaction with TLR-4 paralleled by the inhibition of anti-tumor T cell response upon TIM-3 ligation. All these data are summarized in Figure 2.

In the same figure, we highlight the potential effects of combined administration of anti-CTLA-4 or anti-TIM-3 antibodies with TTX-080 (anti-HLA-G antibody). Indeed, the up-regulation of TIM-3 and CTLA-4 on T lymphocytes driven by HLA-G may be dampened by the inhibition of HLA-G/ILT2 interaction, thus increasing the re-activation of T cell responses against tumor cells.

Since tumor-derived HMGB-1 may inhibit anti-tumor T cell responses through the ligation of TIM-3 on T cells and the up-regulation of HLA-G on tumor cells, an inhibitor of HMGB-1 may be tested in combination with anti-HLA-G antibodies. In this regard, dociparstat (CX-01), a pharmacological inhibitor of HMGB1/TLR4 interaction, has been recently tested in few phase I/II clinical trials conducted on patients with acute myeloid leukemia, with limited side effects (www.clinicaltrials.gov accessed on 30 January 2022). So far, dociparstat showed limited efficacy as monotherapy, but we may speculate that its combined administration with different anti-TIM-3 antibodies, already approved for clinical purposes, plus anti-HLA-G antibodies may result in increased anti-tumor efficacy.

### 2.3. TIGIT

T cell immune receptor with immunoglobulin and ITIM domains (TIGIT) is a surface receptor, originally identified by informatic analysis, that forms a pathway similar to that of CTLA-4/CD28 binding three different ligands that are CD155, CD112, and CD113 all involved in cell adhesion and polarization, tissue organization, and herpes and poliovirus binding. Recently, it has been defined as an emerging IC [85,86].

Limited information is available regarding possible interactions between HLA-G and TIGIT. Martínez-Canales et al. analyzed patients with triple negative breast cancer and found a panel of genes that are up-regulated, and that includes HLA-G and TIGIT. Interestingly, the up-regulation of these two genes, along with HLA-C and -F, is associated with a better outcome of patients in terms of relapse-free survival and overall survival [87]. Although this study demonstrated a paired up-regulation of HLA-G and TIGIT in cancer patients, no evidence of direct interactions between these two molecules has been provided. Furthermore, since the increased expression of HLA-G and TIGIT correlated with a better clinical outcome of patients, these data suggested that, at least in this cohort of patients, these two molecules are not involved in the inhibition of anti-tumor T cell response.

### 2.4. IDO

Idolamine 2,3-dioxygenase (IDO)-1 is a tryptophan catabolic enzyme that has been recently classified as IC [88,89]. It exerts immune inhibitory properties through the metabolic depletion of tryptophan and/or accumulation of kynurenine, leading to the induction and activation of immune-suppressive cells (i.e., regulatory T cells and MDSC, paralleled by inhibition of T and NK effector functions. In the tumor microenvironment, this immune escape mechanism is potentiated by the release of pro-inflammatory cytokines, such as IFN-γ, IL-1, IL-6, and TGF-β1, that in turn up-regulate IDO-1 [88,89]. IDO-1 is expressed in many primary and metastatic tumors such as lung [90], breast [91], brain [92], and gliomas [93], and it is now considered as a target for potential immunotherapy.

Several studies reported a concomitant over-expression of HLA-G and IDO in different human tumors. The expression of HLA-G and IDO has been detected in endometrial cancer, with an increased expression of the two molecules in tumors as compared to surrounding normal tissue [94]. In addition, HLA-G expression was increased in high-grade tumors as compared to those in low-grade, thus suggesting a role of this molecule in the progression of the disease. In contrast, IDO expression was not related to tumor progression and, in addition, the concomitant expression of the two molecules showed no significant difference with cases showing single HLA-G or IDO expression [94]. Thus, IDO and HLA-G immunosuppressive functions in endometrial cancer are not directly related.

Similarly, patients with vulvar squamous cell carcinoma (VSCC) showed a higher expression of HLA-G and IDO in cancer tissue than in surrounding normal tissue [95]. HLA-G expression was strongly associated with disease progression, whereas IDO expression correlated only with the extranodal spread of tumor. The combined high expression of HLA-G and IDO, along with HLA-E, correlated with tumor size and invasion depth. More importantly, the clinical outcome of patients with a high expression of HLA-G or IDO was worse than those with a low expression of these molecules. Furthermore, patients showing a combined high expression of the two molecules, along with high HLA-E expression, display the worst clinical outcome, with very low survival rates [95]. Thus, in subjects affected by VSCC, HLA-G, and IDO, immune-suppressive functions are correlated, and the two molecules may co-operate in the inhibition of anti-tumor immune response, worsening the clinical outcome of patients.

To date, 101 clinical trials using IDO inhibitors in combination with other chemotherapeutic agents are active for cancer patients (www.clinicaltrials.gov accessed on 30 January 2022). Given the correlation between IDO and HLA-G in different human tumors, anti-HLA-G antibodies may be used in combination with these inhibitors, such as Indoximod and Epacadosat, to further increase their effect on the rescue of anti-tumor immune response.

## 3. HLA-G and Other IC in Pediatric Solid Tumors

The role of IC molecules has been described in different pediatric solid tumors, accounting for approximately 50% of cancers in childhood, that include central nervous system tumors (i.e., gliomas and glioblastomas) and extra-cranial solid tumors such as neuroblastoma (NB), Wilms tumor, and sarcomas (e.g., Ewing sarcoma) [64,96]. Of these IC, the most studied are IDO, PD-1, and CTLA-4 that are implicated in the progression of glioma and glioblastoma [97] and, for this reason, have been considered as potential targets for immunotherapy. In this context, different antibodies against CTLA-4 and PD-1 have been developed for clinical purposes and used for immunotherapy in pediatric patients with glioblastoma [98,99]. Indeed, more than 50 clinical trials based on PD-1 blockade are currently active for glioma patients (www.clinicaltrials.gov accessed on 30 January 2022).

The expression of HLA-G and its role in cancer progression has been addressed in gliomas [100,101], glioblastoma [102,103], and Ewing sarcoma [104,105,106], whereas limited information is available on meningioma [107], retinoblastoma [108], and other sarcomas [73].

Our group extensively studied HLA-G as a tumor escape mechanism in NB, which is the most common pediatric extra-cranial solid tumor in childhood [109]. We reported that HLA-G, either on the cell surface or soluble, is expressed in NB cell lines, and NB patients at diagnosis showed an increased level of serum sHLA-G compared to healthy controls. Of note, the sHLA-G in the serum may be shed by tumor cells as well as by monocytes, where sHLA-G release is induced by NB cells. The role of HLA-G as IC and in the progression of NB is further supported by the following findings: first, sHLA-G from patients’ sera inhibited NB cell lysis mediated by NK cells and specific cytotoxic T lymphocytes; second, high serum levels of sHLA-G correlated with a worse prognosis of NB patients [46].

Another interesting finding is the demonstration that metastatic NB cells in the BM of patients expressed high levels of HLA-G. Since primary NB cells tested negative for HLA-G expression, we hypothesized that the metastatic spread of NB cells may be paralleled by the de novo expression of HLA-G on tumor cells [47]. In support of this hypothesis, we found higher sHLA-G levels in BM plasma samples from NB patients with metastatic disease than in samples from those with localized tumors [110]. All these data highlighted the role of HLA-G as novel IC in NB, where the expression and function of other IC molecules has been already demonstrated. In particular, PD-1 expression on NB cells inhibits anti-tumor immune responses [111] and may limit the efficacy of therapies based on anti-GD2 CAR T cells [112,113]. Furthermore, preclinical studies demonstrated the therapeutic efficacy of PD-1 blockade alone [114,115] or in combination with anti-CTLA-4 vaccination [116] in NB animal models. NB cells also express the B7-H3 (CD276) antigen, which is a glycoprotein that belongs to the B7 family of molecules that has been described as an IC molecule in the last years in different solid tumors [117,118]. B7-H3 is expressed on metastatic NB cells from patients, and its pro-tumoral effect is related to the inhibition of NK-mediated tumor cell lysis [119]. To date, eight clinical trials are currently treating NB patients with antibodies against IC molecules: three of them are using Nivolumab, Dostarlimab or Pembrolizumab (anti-PD-1 antibodies), four of them are based on the administration of Ipilimumab (anti-CTLA-4 antibody), and the last one is employing Enoblituzumab (anti-B7-H3 antibody) (www.clinicaltrials.gov accessed on 30 January 2022).

All these data on pediatric solid tumors underline that HLA-G and other IC molecules expressed on tumor cells and in the tumor microenvironment are actively involved in tumor progression, in particular in gliomas, Ewing sarcoma, and NB. Limited information is available regarding possible interactions and/or correlations between HLA-G and other IC molecules in pediatric settings. Of note, the overall survival of high-risk NB patients is still poor, and novel therapies are urgently needed. In this line, combined protocols targeting HLA-G and other IC molecules such as PD-1, CTLA-4, or B7-H3, along with standard chemotherapy, may represent a novel therapeutic approach.

## 4. Conclusions

Several studies in the last years suggested that HLA-G is part of an immunosuppressive loop that involved other IC molecules, in particular PD-1 and CTLA-4. The interaction of these IC molecules and their ligands with HLA-G/ILT2 axis may be relevant for the induction of a tolerogenic microenvironment, which in turn leads to the escape of tumor cells from the control of the immune system. Thus, novel immunotherapeutic strategies aimed at blocking the interaction of PD-1 and CTLA-4 with their ligands, paralleled by the targeting of HLA-G/ILT2 interaction, may greatly enhance the rescue of anti-tumor immune response.

In this view, a prospective clinical trial (GEIA study) is currently recruiting patients with solid tumors receiving anti-PD-1/PD-L1 immunotherapy alone or in combination with anti-CTLA-4. The goal of this study is to determine whether the expression of HLA-G on tumor, along with soluble HLA-G production, may affect the clinical response of patients to immunotherapy, in terms of tumor response rate and progression-free/overall survival (www.clinicaltrials.gov accessed on 30 January 2022). This study will shed a light on the role of HLA-G in the modulation of clinical response to anti-PD-1/PD-L1 immunotherapy combined to anti-CTLA-4, thus paving the way to design novel therapeutic protocols that may include anti-HLA-G or anti-ILT2 antibodies (Table 1).

In conclusion, the combination of anti-HLA-G antibodies with antibodies targeting PD-1, CTLA-4, TIM-3, or with inhibitors of IDO may represent a novel strategy to achieve the full rescue of immune responses against tumors, thus improving the clinical outcome of patients enrolled. However, the side effects of antibodies targeting IC must be taken into account, considering that these antibodies promote a broad activation and expansion of T cells. Activated T cells may infiltrate the most organs, causing a wide range of immune-related adverse events (irAEs), and these can affect virtually any organ, with different grades of toxicity. Furthermore, it has been demonstrated that irAEs are more frequent and severe when antibodies against IC are administered in combination [120]. Several guidelines for the management of irAEs are available for clinicians, regarding the type of immunosuppressive drugs to use and the duration of treatment based on the severity of the irAEs [120]. In addition, it is mandatory to determine (i) the optimal agent for combined therapies based on antibodies targeting IC and the (ii) dose and timing of administration, to limit the adverse events.

## Figures and Tables

**Figure 1 ijms-23-02925-f001:**
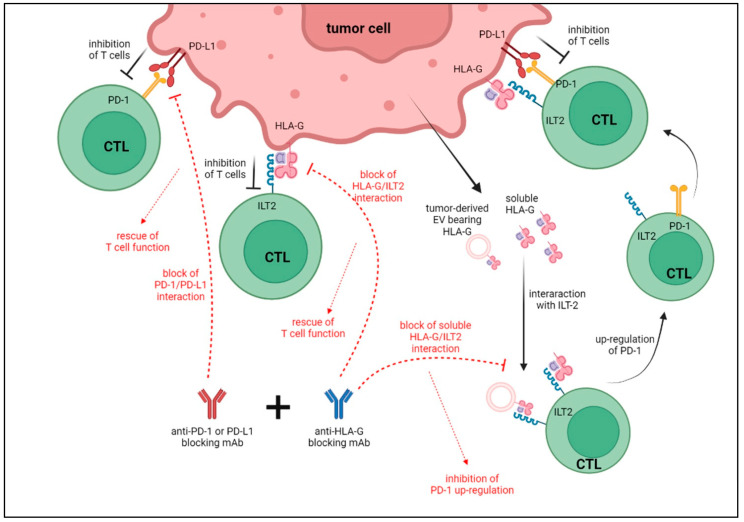
Schematic representation of the interactions between HLA-G and PD-1/PD-L1 and possible effects of a combined administration of anti-PD-1/PD-L1 and anti-HLA-G blocking antibodies.

**Figure 2 ijms-23-02925-f002:**
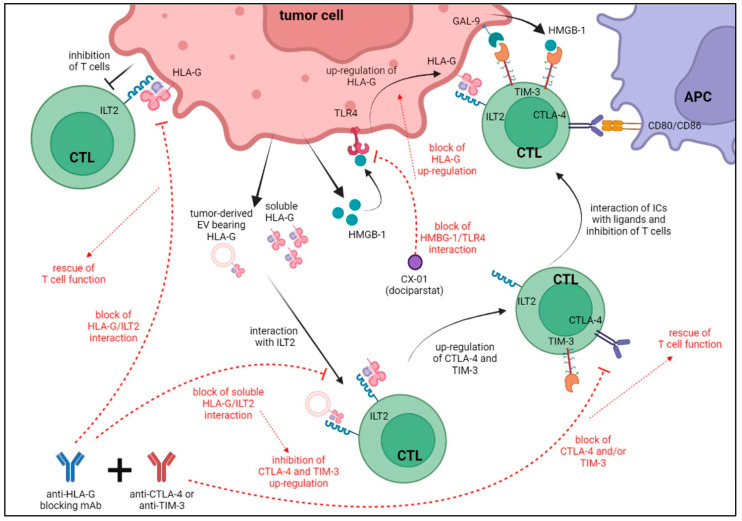
Schematic representation of the interactions of HLA-G with CTLA-4/TIM-3 and their ligands. Possible effects of the administration of anti-CTLA-4 and/or anti-TIM-3 in combination with anti-HLA-G blocking antibodies are also shown.

**Table 1 ijms-23-02925-t001:** Description of clinical trials based on the combination of antibodies targeting HLA-G and other immune checkpoints.

ID	Title	Phase	Aim of the Study
NCT04485013	TTX-080 HLA-G Antagonist in Subjects With Advanced Cancers	Ia/Ib	Safety and tolerability of TTX-080, a fully human mAb against HLA-G, alone or in combination with pembrolizumab (anti-PD-1 mAb) or cetuximab (EGFR inhibitor) in patients with Head and Neck Squamous Cell Carcinoma (HNSCC), colorectal cancer (CRC), non-small cell lung cancer (NSCLC) and triple negative breast cancer (TNBC)
NCT04991740	A Study of JNJ-78306358 in Participants With Advanced Stage Solid Tumors	I	Safety and tolerability of JNJ-78306358, a bispecific antibody binding to CD3 on T cells and HLA-G on cancer cells, in patients with renal cell carcinoma (RCC), ovarian cancer, CRC), lung adenocarcinoma, endometrial cancer, and pancreas cancer
NCT04300088	A Prospective Study of the Relevance of the HLA-G Immune Checkpoint in Cancer Immunotherapy (GEIA)	-	The impact of HLA-G tumor expression (evaluated by immunohistochemistry) on tumor response rates (evaluated with iRECIST) in patients with NSCLC, RCC, urothelial carcinoma and other tumors treated with immunotherapy against PD(L)1 and CTLA4

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
