# Peer review of "HLA-G and Other Immune Checkpoint Molecules as Targets for Novel Combined Immunotherapies"

_ijms, 2022, doi:10.3390/ijms23062925_

Round 1

Reviewer 1 Report

Nothing to add.

Author Response

We thank the Reviewer for the appreciation of our manuscript.

Reviewer 2 Report

The authors present two mechanisms of immunosuppression by HLA-G. 1) inhibition of CTL function via HLA-G-binding to ILT-2 on T cells, 2) Binding of tumor-derived, soluble HLA-G and Envelope-HLA-G to ILT-2 on CTLs to increase expression of PD-1 and HMGB-1 on the CTLs.

Figures 1 and 2 are excellent because they give readers clear images of the concept of immunosuppression by HLA-G and the combination antibody therapy targeting it.

What we believe is lacking in current MS is 1) information on intracellular signaling of ILT-2 downstream and 2) a table summarizing clinical trials of antibody therapies targeting HLA-G and other checkpoints.

With these, this review will be more informative and helpful to the reader in understanding "HLA-G as an immune checkpoint mechanism."

Minor points:

1)

CD69 is not an exhaustion marker, but an early activation marker.

2)

Correct the misspelling. For example, P.7 line 299,  binging → binding

Author Response

The authors present two mechanisms of immunosuppression by HLA-G. 1) inhibition of CTL function via HLA-G-binding to ILT-2 on T cells, 2) Binding of tumor-derived, soluble HLA-G and Envelope-HLA-G to ILT-2 on CTLs to increase expression of PD-1 and HMGB-1 on the CTLs. Figures 1 and 2 are excellent because they give readers clear images of the concept of immunosuppression by HLA-G and the combination antibody therapy targeting it. What we believe is lacking in current MS is 1) information on intracellular signaling of ILT-2 downstream and 2) a table summarizing clinical trials of antibody therapies targeting HLA-G and other checkpoints.With these, this review will be more informative and helpful to the reader in understanding "HLA-G as an immune checkpoint mechanism."

We thank the Reviewer for these comments and suggestions. We have added the description of intracellular signaling of ILT2 and also of ILT4 and NKG2D (page 2). There are few clinical trials based on the use of antibodies targeting HLA-G, and only two of them are based on the combined administration of antibodies targeting HLA-G and other immune checkpoints. One additional study is investigating the impact of HLA-G expression on the clinical outcome of patients treated with anti-PD-1 and anti-CTLA-4 antibodies. Despite of the limited number of available clinical trials, we have summarized them in the new Table 1, as suggested.

Minor points:

1) CD69 is not an exhaustion marker, but an early activation marker.

We agree with the Reviewer that CD69 is an early activation marker. However, recent studies demonstrated that CD69 may also represent a marker of exhaustion in T cells (Mita et al, Int Immunol 2018). In the cited reference (#79, Dumont et al, Cancer Immunol Res 2019) CD69 has been evaluated as exhaustion marker in ILT2+ and PD1+ T cells. Accordingly, the authors demonstrated that CD69 is expressed only in PD1+ T cells.

2) Correct the misspelling. For example, P.7 line 299,  binging → binding

We apologize for these errors. The manuscript has been checked and misspellings have been corrected.

Reviewer 3 Report

In this review, the authors have provided an overview of combinatorial approaches involving HLA-G and other immune checkpoint molecules in cancer. The topic of the review is of high importance and is nicely written to provide critical insight into this rapidly evolving field of checkpoint blockade therapeutics. Overall, the flow of the review is good and is easy to understand for the readers. However, this reviewer has some major and minor points that may further strengthen the manuscript.   

Major points:

  • Authors should include a table with an overview of combinatorial approaches involving HLA-G axis and other IC molecules – summarizing important studies (preclinical and clinical) and specifying a combination of HLA-G and a specific IC axis, name/target of antibodies, cancer type, outcome and reference
  • Authors should at the end discuss limitations and challenges that they may envision in pursuing HLA-G and IC checkpoint combinatorial strategies – including possible side effects in patients (irAE)  
  • Line 114 – 119: “During the elimination phase, HLA-G functions with paralleled mechanisms, through the ILT2/ILT4 signaling, that involves the inhibition of i) proliferation of T and B lymphocytes, ii) cytotoxicity of NK and T cells, iii) phagocytic activity of neutrophils, and iv) function of DC[55-59]. Furthermore, HLA-G dampens the secretion of pro-inflammatory cytokines[26] and maybe up-regulated, thus providing an additional immune escape mechanism to cancer cells[12, 60].“ ----- This is confusing as these interactions are actually part of the equilibrium/escape phase of immunosurveillance as opposed to the elimination phase. Please revise it to clarify this point.  Line 111 - 127: Or rather, authors may remove the role of HLA-G in each of the different phases (elimination, equilibrium, and escape) but just summarize at the end of the whole paragraph that HLA-G plays an important role in promoting tumor immune escape. In the same paragraph, authors should include the names of the cancers where and when these HLA-G functions in the context of immune escape are noted.

Minor points:

  • Figures: Figures 1 and 2 appear blurred. Authors should replace the figures with high-resolution images. Also, please remove the writing “Figure 1” and “Figure 2” inside the figures.
  • Line 28 or Line 75: A sentence on some common pathological conditions apart from cancer where HLA-G plays a disease-modifying role will be apt at either Line 28 or Line 75
  • Line 73- 76: This paragraph can be merged with the paragraph above
  • Line 73: Authors should write briefly about the physiological functions of HLA-G in adults in the tissues mentioned.
  • Line 144: Please also include tumor cells as cells expressing IC ligands

Language related minor points:

  • Line 26: …molecules that also include also HLA….
  • Line 27: HLA-G, among which it is the most characterized --- Among these, HLA-G is the most characterized…
  • Line 56: The addition of “and is” makes it clear that ILT4 is present on monocytes, neutrophils, DC…but also β2M-free HLA-G1 and HLA-G5, and is mainly present on monocytes, neutrophils, DC…
  • Line 77: In the context of pathological conditions, cancer is always the major focus of interest: The tone of this sentence sounds negative (“cancer is always the major focus..”). Please revise it to support why cancer is the major focus of interest and that it is rightfully so.
  • Line 81-83: However, although an enhanced sHLA-G (i.e., HLA-G5 and shed HLA-G1) plasma levels has been observed, no clear correlation was established between HLA-G and unfavorable clinical outcome in these tumors. Please provide citations.
  • Line 122-123: by promoting the expansion of regulatory T cells and the secretion of an-
  • Line 132: target to be used also for chimeric antigen receptor (CAR) therapy development.
  • Line 206: CD127, and highly expressed KIRG1, perforin and… -- KLRG1?
  • Line 227: In details, the block of --- In detail, the blocking of..
  • Line 266: TIM-3, similarly to CLA-4, belongs .. – CTLA-4
  • Line 352 - accounting for approximately 50% of pediatric cancers in pediatric age
  • Line 369 - sHLA-G in the serum may be released by tumor cells as – Shed instead of released may be better in this context.
  • Line 417 - This study will shed a light on the role (should be – shed light on the role)

Author Response

In this review, the authors have provided an overview of combinatorial approaches involving HLA-G and other immune checkpoint molecules in cancer. The topic of the review is of high importance and is nicely written to provide critical insight into this rapidly evolving field of checkpoint blockade therapeutics. Overall, the flow of the review is good and is easy to understand for the readers. However, this reviewer has some major and minor points that may further strengthen the manuscript.  

Major points:

 Authors should include a table with an overview of combinatorial approaches involving HLA-G axis and other IC molecules – summarizing important studies (preclinical and clinical) and specifying a combination of HLA-G and a specific IC axis, name/target of antibodies, cancer type, outcome and reference

We thank the Reviewer for this suggestion. We now added Table 1, where two available clinical trials based on the combination of antibodies targeting HLA-G and other IC are described. In addition, we described one clinical trial where the clinical outcome of patients treated with anti-PD-1 antibodies is correlated with the expression of HLA-G. However, all these studies are currently recruiting patients. Thus, the outcome of these trials is so far not available.

Authors should at the end discuss limitations and challenges that they may envision in pursuing HLA-G and IC checkpoint combinatorial strategies – including possible side effects in patients (irAE) 

We thank the Reviewer for this comment and we totally agree with Him/Her. To cope with this criticism, we have added a paragraph in the discussion regarding this issue (page 10).

Line 114 – 119: “During the elimination phase, HLA-G functions with paralleled mechanisms, through the ILT2/ILT4 signaling, that involves the inhibition of i) proliferation of T and B lymphocytes, ii) cytotoxicity of NK and T cells, iii) phagocytic activity of neutrophils, and iv) function of DC[55-59]. Furthermore, HLA-G dampens the secretion of pro-inflammatory cytokines[26] and maybe up-regulated, thus providing an additional immune escape mechanism to cancer cells[12, 60].“ ----- This is confusing as these interactions are actually part of the equilibrium/escape phase of immunosurveillance as opposed to the elimination phase. Please revise it to clarify this point.  Line 111 - 127: Or rather, authors may remove the role of HLA-G in each of the different phases (elimination, equilibrium, and escape) but just summarize at the end of the whole paragraph that HLA-G plays an important role in promoting tumor immune escape. In the same paragraph, authors should include the names of the cancers where and when these HLA-G functions in the context of immune escape are noted.

We agree with this comment, We have followed the Reviewer’s suggestion and we have changed the paragraph (page 3) summarizing only general effects of HLA-G on anti-tumor immune response.

 Minor points:

Figures: Figures 1 and 2 appear blurred. Authors should replace the figures with high-resolution images. Also, please remove the writing “Figure 1” and “Figure 2” inside the figures.

Resolution of Figures has been increased and the writings have been removed, as requested.

Line 28 or Line 75: A sentence on some common pathological conditions apart from cancer where HLA-G plays a disease-modifying role will be apt at either Line 28 or Line 75

We have now added a sentence with new references in former line 75 (page 2).

Line 73- 76: This paragraph can be merged with the paragraph above

We modified the text as requested (page 2).

Line 73: Authors should write briefly about the physiological functions of HLA-G in adults in the tissues mentioned.

We added a short paragraph with this information.

Line 144: Please also include tumor cells as cells expressing IC ligands

We added tumor cells as requested.

Language related minor points:

 Line 26: …molecules that also include also HLA….

Line 27: HLA-G, among which it is the most characterized --- Among these, HLA-G is the most characterized…

Line 56: The addition of “and is” makes it clear that ILT4 is present on monocytes, neutrophils, DC…but also β2M-free HLA-G1 and HLA-G5, and is mainly present on monocytes, neutrophils, DC…

Line 77: In the context of pathological conditions, cancer is always the major focus of interest: The tone of this sentence sounds negative (“cancer is always the major focus..”). Please revise it to support why cancer is the major focus of interest and that it is rightfully so.

Line 81-83: However, although an enhanced sHLA-G (i.e., HLA-G5 and shed HLA-G1) plasma levels has been observed, no clear correlation was established between HLA-G and unfavorable clinical outcome in these tumors. Please provide citations.

Line 132: target to be used also for chimeric antigen receptor (CAR) therapy development.

Line 206: CD127, and highly expressed KIRG1, perforin and… -- KLRG1?

Line 227: In details, the block of --- In detail, the blocking of..

Line 266: TIM-3, similarly to CLA-4, belongs .. – CTLA-4

Line 352 - accounting for approximately 50% of pediatric cancers in pediatric age

Line 369 - sHLA-G in the serum may be released by tumor cells as – Shed instead of released may be better in this context.

Line 417 - This study will shed a light on the role (should be – shed light on the role)

We have corrected all these sentences, as requested.

 Line 122-123: by promoting the expansion of regulatory T cells and the secretion of an-

This is due to the format of the Journal. The word is “an-ti inflammatory” and continues in the next line.

Round 2

Reviewer 3 Report

Dear authors, 

Thank you for providing the revisions. The manuscript is well revised and all my concerns have been addressed satisfactorily. Just a couple of very minor points as below.

1) Line 439: targeting IC must be taken into account

2) Table appears to be blurred. 

Congratulations and all the best!